# Research on the Impact of the Input Level of Digital Economics in Chinese Manufacturing on the Embedded Position of the GVC

**Guangwei Rui [1,2] and Menggang Li [2,*]**

[1] School of Economics and Management, Beijing Jiaotong University, Beijing 100044, China; 20113009@bjtu.edu.cn

[2] Beijing Laboratory of National Economic Security Early-Warning Engineering, Beijing Jiaotong University, Beijing 100044, China

[*] Correspondence: morganli@vip.sina.com

**Abstract:** With the development and application of digital technology, the digital economy industry has gradually become the new vitality of China's economic growth, and it has also become a vital driving force to promote a change in the GVC division of the manufacturing industry. This paper takes the embedded position of the GVC in the Chinese manufacturing industry as the research object, places the input level of the digital economy in the manufacturing industry into the analysis framework of the influence of its embedded position in the GVC, puts forward the theoretical mechanism of the influence of the input level of the digital economy on the relative breadth and height of its embedded position in the GVC, and explores the influence of the digital economy on the embedded position of the GVC in the Chinese manufacturing industry from the two levels of relative height and width. Through regression analysis, an intermediary effect test, and threshold regression of panel data, the study found that (1) improving the input level of the digital economy in manufacturing will positively affect the relative height and breadth of the GVC embedding position. (2) The improvement of the input level of the digital economy plays a role through two mechanisms: improving the innovation efficiency of the manufacturing industry, and improving the asset utilization efficiency of the manufacturing industry. The relative height and breadth of the embedded position of the GVC can be promoted through innovative efficiency channels. The captive allocation efficiency channel can promote the relative breadth of the embedded position of the GVC. (3) The influence of the input level of the digital economy on the relative breadth and height of the embedded position of the GVC presents a threshold effect with the technical level, and the influence on the relative height presents a threshold effect with the capital level. By clarifying the influence of the digital economy's input level on the embedded position of the GVC, some suggestions can be taken to promote the manufacturing industry to move to a high-value acquisition position in the GVC division. Construction can be strengthened from the following aspects: improving the application level of digital technology in the manufacturing industry, strengthening the construction of digital infrastructure, and promoting the innovation system and industrial ecology led by digital technology.

**Keywords:** digital economics; the global value chain; the manufacturing industry

## 1. Introduction

Since the 1960s, the division of labor in the value chain has gradually become popular. With the diversification and deepening of production cooperation among countries around the world, the global production network spawned by the division of labor within the product has continued to develop, and the corresponding GVC (global value chain) has also extended. However, in recent years, the international political and economic situation has become more volatile, global economic growth has been affected by many "black swan" events, uncertainty has increased significantly, and the risk of a "broken

chain" in the division of labor in the value chain has increased. The improvement of the embedded position of the GVC is of great significance to various economies, especially emerging economies. The resistance to technology flow in the world market is gradually increasing, and the original "learning by doing" promotion path for emerging markets has increasingly narrowed. In this context, countries are actively promoting the status of the division of labor in the value chain, participating under the GVC system, and continuously improving their position, which has become a meaningful way to promote the expansion of international trade and economic development of economies [1]. With the increasingly tight international division of labor, the status of the GVC has also become an important source and measurement standard of global structural power and "positional power" [2,3].

At the same time, a new generation of information technology has promoted the rise and development of the digital economy in the global scope, brought revolutionary changes in industrial technology routes and breakthrough innovations in business models, and gradually become an emerging industry that all economies in the world are focusing on. In addition to digital infrastructure such as communication equipment and network signal service, the digital economy has penetrated all aspects of social development, and its influence on the manufacturing industry is equally apparent. In the international division of labor system, the input of the digital economy realizes the virtualization of manufacturing factors, transactions, and transmission, as well as the digital delivery process of goods trade, accelerates the transmission speed of information across regions, and improves the efficiency of coordinated distribution of factors across areas. In its development, the digital economy has gradually become a new strategic economic development paradigm that leads to a new technological revolution and promotes industrial transformation. It is also an essential strategic starting point for winning the initiative of world science and technology competitions and a significant strategic opportunity for the new round of industrial revolution of the economy [4].

In this context, the digital economy-related industries in significant economies have developed rapidly and gradually become essential to economic operations. In 2021, the digital economies of the United States, China, Germany, Japan, Britain, and France exceeded 1 trillion US dollars. Among them, the digital economy in the United States reached 15.3 trillion US dollars, with a growth rate of 12.50%; The scale of China's digital economy is 7.1 trillion US dollars, with a growth rate of 31.48%; The scale of the digital economy in Germany is 2.9 trillion US dollars, with a growth rate of 14.17%. The scale of the digital economy in various countries has developed well. The growth rate is relatively fast, which is higher than the national GDP growth rate and an essential engine of current economic growth. At the same time, governments worldwide have explicit policies to support the development of related industries. For example, the European Union issued the "European Data Strategy" to further promote the integration of the European data market. The UK released the "UK digital strategy" in 2021, focusing on improving six significant areas of digital economy development and promoting the development of the digital economy in the UK to be more inclusive, competitive, and innovative. Germany updated "Digital Strategy 2025" to promote the sustainable development of digital technology, infrastructure and equipment, innovation and digital transformation, and personnel training. To sum up, the digital economy industry has become an essential part of the layout of various countries.

In the international division of labor, the value distribution based on differences in the development foundation and technological level of economies still exists. After increasing digital investment, high-tech economies rely on the "first-mover advantage" of existing digital platforms and digital elements to get more value assignments. At the same time, as an emerging economy and the largest developing country, China is actively seeking to move the embedded position of the GVC to a high-value position in the division of labor in the GVC. China is an important manufacturing country in the world market. It has all 39 industrial classifications in the International Standard Industrial Classification of Economic Activities. It has maintained its status as the highest-value manufacturing export country for more than 10 years, with a large scale and sound system. By 2021, the

added value of China's manufacturing industry will be 31.38 trillion CNY, accounting for a large proportion of international trade in manufacturing. At the same time, China's digital technology is developing rapidly and has a high degree of coverage in the country's financial operations. With the continuous investment in digital technology and digital facilities in recent years, the scale and stock have increased significantly, and it is now in a period of vigorous development. In 2022, the scale of China's digital economy will be 50.2 trillion CNY, a nominal increase of 10.3% year-on-year, 3.4 percentage points higher than the nominal GDP growth rate of the same period, and accounting for 41.5% of the GDP. There is a "U"-shaped relationship between China's position embedded in the GVC and the degree of transformation and upgrading of the manufacturing industry [5], and the profitable industries are mainly concentrated in traditional resource-intensive and labor-intensive sectors. It is a large-scale industry with room for improvement and demand for improvement. Against this background, there is a solid practical foundation to study the effect of digital economy investment on the embedded GVC position of Chinese manufacturing and clarify its influence mechanism. Therefore, this paper aims to explore the impact of digital economy investment on the embedding position of GVC in the Chinese manufacturing industry and clarify its influence mechanism. Through the exploration of the above problems, it is helpful to describe the influence effect of digital economy input level on the national manufacturing industry from the perspective of the embedding position of the GVC and put forward a favorable reference for the development direction of the manufacturing industry.

This paper mainly uses literature and empirical research methods and develops according to the level of "theory-mechanism-demonstration-application". In empirical research, panel data is used to construct a panel regression model for research, and a mediation effect model and a threshold regression model are constructed for further analysis. The remainder of this article is as follows: Section 2 returns to the existing literature and clarifies the theoretical basis of the research; Section 3 analyzes the theoretical mechanism, improving the embedded position of the GVC and the embedded value of the GVC from the digital economy In addition, from the two perspectives of innovation level and capital utilization efficiency, the mechanism analysis of the promotion of the embedded position of the GVC is proposed; Section 4 describes the experimental design and the measurement methods for the digital economy, and the embedded GVC degree of China is analyzed, while also describing the actual situation; Section 5 presents empirical tests. First, the impact of China's digital manufacturing economy input level on the GVC by regression is evaluated. Furthermore, analyses of the industry heterogeneity, mechanism, capital, and threshold effect of technological innovation are tested; Section 6 provides the conclusion and suggestions.

## 2. Literature Reviews

### 2.1. Theoretical Connotation and Measurement of Digital Economy

Regarding the concept and theoretical connotation of the digital economy, Don [5] (1994) was the first to define the idea of the digital economy accordingly. Since the 21st century, with the development of ICT (information and communications technology), the digital economy has been defined as economic activities in which goods and services are traded in digital form [6], including the extensive application of the financial system of technology, including infrastructure, e-commerce, and electronic transactions [7]. With the increasing participation of digital technology in international economic activities, the status of digital assets as a factor of production has begun to be generally recognized, and discussions on the concept and theoretical connotation of the digital economy have become more in depth. In 2016, the "G20 Digital Economy Development and Cooperation Initiative" proposed at the G20 summit mentioned that the digital economy regards digital knowledge and information as the key production factors, including the primary and integration parts of the digital economy [8]. At the same time, with the development of digital technology, the concepts of data information and data assets have been valued, and

the flow of data information has also been summarized as digital economic activities [9]. From the perspective of new financial forms, data information is the key element of the digital economy [10,11]. Furthermore, according to the investment in digital technology, the digital economy can be classified into industries directly related to digital technology, namely, "digital industrialization", and the economic form produced after the integration of digital technology, namely, "industrial digitalization" [12]. On this basis, the input level of the digital economy in the industry is the use of digital technology in sectors directly related to and integrated into digital technology industries in industrial development.

With the development of digital technology, the critical role of the digital economy in economic operations has gradually emerged. As a factor endowment, the digital economy promotes economic growth through the integration of production factors [13,14], which can be reflected explicitly in the promotion of total factor productivity, reducing transaction costs, and promoting the effect of technological innovation [15].

Since digital technology participates in all aspects of economic operation, it shows "substitution", "permeability", and "synergy" [16–18]. The characteristics of the digital economy make it difficult to measure its growth, which has become the focus of academic discussions. According to its development characteristics and evolution path, the digital economy is generally divided into primary digital sectors, digital economy integration sectors, and alternative digital economy sectors. In terms of specific measurement methods, two types are commonly used at present: digital economy measurement by constructing a digital economic growth index system [19,20]; for the scope of the digital economy, using tools such as input–output tables to calculate the added value of the "digital industrialization" and "industrial digitization" parts [21,22]. On this basis, for the calculation of the industry's digital economy input, on the basis of the input–output table, the ratio of service input to total input in the manufacturing industry can be used to quantify the servitization of manufacturing input [23,24]. Furthermore, with the digital economy industry definition, the manufacturing industry's digital input level can be calculated as a function of the proportion of the total output of the digital industry intermediate products used in manufacturing production.

### 2.2. Research on Embedded Location of the GVC

With the gradual improvement of industrial differentiation, the production division has gradually taken shape, and the value chain concept has steadily taken form. This concept was initially portrayed from the perspective of enterprises, and the process of creating value by enterprises was subdivided into several relatively independent individuals. The value created by these production and business activities was formed in series, including a "value chain" [25]. On this basis, with the increasingly close industrial ties in the world, the concept of a global value chain (GVC) has gradually formed, emphasizing the network organization formed by connecting all links of production activities to realize the value of goods or services worldwide [26,27]. From the perspective of the GVC division, the embedding position of the economic subject in the GVC reflects the ability of an economic subject to obtain value in the GVC. When the economic issue is close to the high-value acquisition position, its ability to acquire value is also improved. From this perspective, the theoretical connotation of upgrading the status of the GVC is reflected in the established global dividend and the improvement of the ability to benefit [28]. Therefore, the driving force of the embedded position of the GVC in the economy toward high-value acquisition can be regarded as the "climbing" of the GVC. According to different perspectives, the shift from an embedded position to high-value addition has several various aspects: the relative height of the GVC based on the length of the GVC, which can be expressed as the size of a country or its departments' interest acquisition ability and the level of division of labor in the GVC [29]; the relative breadth of the GVC based on the description of the embedded depth of the GVC, which can be expressed as the proportion range involved in the production of the world economy.

Several aspects can be used to discuss the driving factors promoting high-value GVC acquisition. First, from the perspective of factor endowments, differences in factor endowment levels among countries determine each country's specialization level [30,31]. Generally speaking, the higher the factor endowment abundance, the higher the production efficiency of industries related to factor endowments, and the more pronounced the comparative advantage of the initiative, which will promote the industry's GVC embedded position to drive to a high-value position [32]. Labor endowment, capital endowment, and technology endowment are all important directions that affect the development of the GVC [33–35]. In addition, innovation is an essential driving factor for the "climbing" of the GVC [36]. For emerging economies, the innovation drive leads to the innovation of domestic intermediate products, realizes import substitution, and then triggers skill-biased technological progress. Under the guidance of this mechanism, innovation drive can promote the GVC of the economy to "climb" [37]. At the same time, from the perspective of the theory of economies of scale, economies of scale determine the level of horizontal specialization and vertical specialization of the industry, which are decisive factors for the country to participate in the international division of production. This reduces the production cost of enterprises and promotes internal learning of enterprises, affecting the international division of labor [38–41]. Lastly, transaction costs can affect firms' decision to embed in GVCs [42,43]. The reduction in transaction costs promotes the trade of intermediate goods, and measures such as trade facilitation conditions and government systems can affect transaction costs, thereby affecting the embedded position of the GVC [44,45].

*2.3. The Impact of the Level of Digitalization on the Embedded Position of the GVC*

The research on the theoretical mechanism of the digital economy's influence on the GVC climb is mainly analyzed from the below-described perspectives.

Firstly, the digital economy can reduce transaction costs and improve the status of the GVC. On the one hand, the digital economy has lowered transportation costs, but the impact of distance still exists. At the same time, retail demand, cultural differences, and highly localized social networks are still important factors influencing online transactions. In addition, the digital economy has dramatically reduced information costs or even approached zero. Digital technology has improved the efficiency of trade transactions, weakened the restrictions on trade space distance, reduced the uncertainty of trade time, and lowered trade thresholds, enabling more enterprises to participate in the division of labor in the global GVC with higher efficiency [46]. By reducing transaction costs, uncertainty and production costs in the international division of labor and trade are reduced, thereby affecting the position of the GVC.

Secondly, the related research demonstrates that the digital economy affects the GVC position by affecting the distribution of the added value of production factors and improving production efficiency. In terms of influencing the distribution of added value, digital technology promotes the standardization of goods and services and enhances the flexibility of the global GVC. The added value obtained by participating in the division of the global GVC is increasingly related to digital technology [47]. At the same time, digital technology also affects the distribution of added value in all links of the global GVC when reconstructing the division of production. Intelligent production processes can obtain more added value, and the added value of R&D, design, and sales links also changes due to digital technology investment [48]. At the level of enterprise development, the application of digital technology can effectively improve the profit level obtained by enterprises [49].

Regarding improving production efficiency, scholars believe that digital technology can increase added value by improving the efficiency of production cooperation. For example, digital technology can improve production efficiency by improving the collaboration of traditional production factors [50], optimizing production processes to enhance the productivity of existing production factors [51], and improving total factor productivity [15]. Digital technology can also promote customized production and improve efficiency by shortening setup, running, and inspection times [52].

Lastly, for emerging economies, the innovation of intermediate products drives the climb of the global GVC [37]. Scholars have studied the digital economy to enhance the status of the GVC by improving innovation efficiency. The digital economy can improve innovation efficiency and promote technological progress. From the perspective of industrial upgrading, the digital economy can promote technological progress and enhance national competitiveness, making it a high-end position in the GVC in the global industrial division of the labor system. The two integrate and promote the development of strategic emerging industries through informatization, networking, and intelligent technologies [53]. In addition, digital technology can significantly promote enterprise R&D investment, product innovation, and process innovation, thereby improving enterprises' technological innovation level. At the same time, this can enhance the innovation spillover effect of foreign direct investment. The threshold regression model shows that the digital economy can also significantly improve the efficiency of regional innovation [54].

### 2.4. Summary of Existing Research

From the above literature review, existing studies have achieved broad research results in describing the status of the GVC and quantifying the development level of the digital economy. The promotion path of the embedded position of the global GVC is discussed, and the influence of the digital economy on it is analyzed accordingly. Compared with the previous literature, this paper can supplement the research in related fields in the following aspects: (1) from the perspective of input and output, this paper quantifies the degree of digitalization of the manufacturing industry through the calculation of objective data; in the embedded position of the GVC, this is a valuable expansion of related research to analyze the GVC length and GVC breadth. (2) Extensive research exists on the digital economy's motivation to change the GVC's embedding position. Because the degree of digitalization of the manufacturing industry and the embedding degree of the GVC itself are still critical research contents, the corresponding empirical tests are relatively limited. This paper adds empirical evidence to the existing research by quantifying the GVC's embedding position and the manufacturing industry's embedding degree. (3) This paper examines the theoretical mechanism of influence from the perspective of reducing transaction costs, improving capital utilization, and innovation efficiency. The threshold effect test analyzes the threshold mechanism of technology and capital levels.

### 3. Analysis of Influence Mechanism

The input level of the digital economy in manufacturing affects the embedded position of the GVC in two ways: innovation efficiency and asset allocation efficiency. The theoretical path is shown in Figure 1.

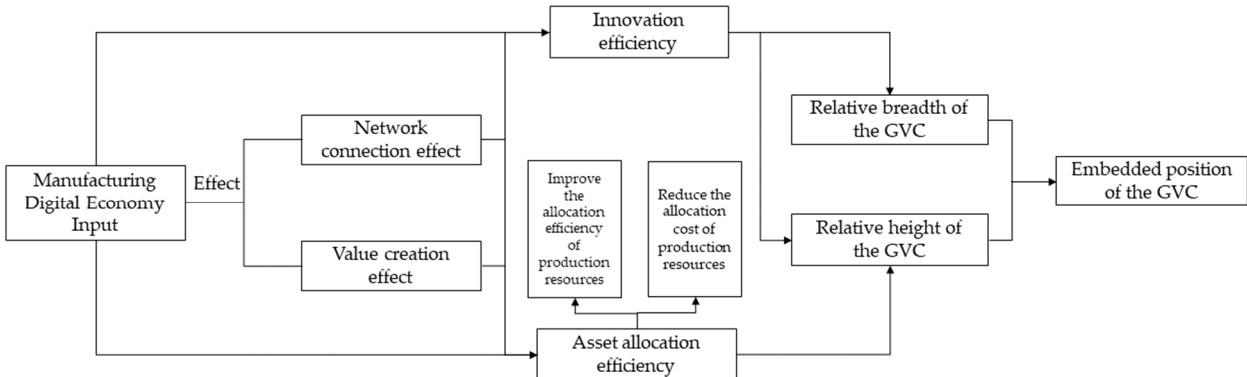

**Figure 1.** Theoretical mechanism conduction.

### 3.1. Analysis of the Effect on the Embedded Position of the Manufacturing Value Chain

The digital economy has the characteristics of "substitution", "permeability", and "synergy", which act on many levels in the process of manufacturing development. With

the improvement of the digital input level of the manufacturing industry, it has an impact on the movement of the embedded position of the GVC of the manufacturing industry through the below-described aspects.

Firstly, the input of the digital economy has a network link effect. From the perspective of enterprises, the wide application of digital technologies such as cloud computing, big data, and artificial intelligence can improve the connectivity between enterprises, thus promoting the integration of enterprises in all production links into the division of labor system of the GVC. In industry, digital technology can choose more advantageous value chains through higher information circulation, thus promoting upstream and downstream enterprises to participate in the production process of value chains more widely and improving the participation degree of the GVC. From this perspective, promoting the digital economy affects the relative breadth of the GVC embedding of the economy through the network link effect.

Secondly, the development of the digital economy shows a value-creation effect on the manufacturing industry. On the one hand, the application of digital technology in the manufacturing industry has promoted the development of digital technology-related industries, and, as a new industry, it has participated in the division of the GVC and increased its participation in the status of the GVC. On the other hand, the digital economy is permeable and engages in all aspects of manufacturing production, thus increasing the added value in all production processes of the GVC. From the show and sales links, in the R&D and design link, through the multi-port characteristics of the digital platform, the original R&D and innovation independently completed by enterprises has evolved into a collaborative innovation mode between the mass group and many enterprises. The capital and technical threshold for the R&D and design of enterprises has been lowered, creating convenience for more enterprises to enter this link. Secondly, in the processing link, the enterprise can change the traditional assembly line into an automatic, intelligent, and flexible thoughtful manufacturing process by introducing automated production equipment. The process can improve the control of the production link and product quality accordingly, and a higher proportion of added value can be obtained in GVC. In the sales process, by building a sales system with a vast platform and massive terminals, enterprises can accurately identify consumers' potential consumption needs and after-sales maintenance needs, formulate sales plans in advance, allocate factor resources efficiently, and increase the added value of sales and after-sales service. From the above two perspectives, the relative position of the GVC will be promoted.

On the basis of the above analysis, the following hypotheses can be proposed:

**H1:** *Improving the input level of the digital economy in the manufacturing industry is conducive to the embedded position of the GVC to a high level.*

**H1.1:** *Improving the input level of the digital economy in the manufacturing industry is conducive to improving the embedded breadth of the GVC.*

**H1.2:** *Improving the input level of the digital economy in the manufacturing industry is conducive to improving the relative height of the GVC embedding position.*

*3.2. Improve Innovation Efficiency*

The impact of technological innovation on the rise of the GVC has been widely investigated and recognized by academic community. According to the phased development model of competitive advantage, the advantage will gradually rise from simple factor promotion to investment promotion and development to innovation promotion. The driving force of position fluctuation from low to high in the GVC transforms pure factor capital into innovative capital [55]. Technological upgrading is a crucial way to climb the embodied position of the GVC for emerging economies. In the early stage of economic growth, we can take advantage of late development through "learning by doing", imitating and absorbing the advanced technology and experience of developed countries, and significantly reducing

the cost and risk of innovation [56]. Through the improvement of the technological level, the improvement of the digital level of the manufacturing industry has an impact on the embedded position of the GVC of the manufacturing. This effect is also reflected in the improvement of the relative position and the breadth of the GVC.

Through the improvement of the technology level, the improvement of the input level of the digital economy in manufacturing impacts the embedded position of the GVC. This impact is also reflected in the improvement of the relative position and breadth of the GVC.

From the perspective relative position of the GVC, the theoretical model of Zheng Jianghuai and Zheng Yu [37] can be used to discuss it. In the embedded position of the GVC, the production of intermediate products in the middle and high stages contains the value of the input of the previous production; thus, there will be a higher price for the formed products. Each production entity's price level is the same at a critical stage. This level is defined as $P_E^*$, the part with the highest value added in the value chain is defined as $P_D^{top}$, and the value added of the product in the initial stage of value is defined as $P_E^0$. Therefore, the ratio of high value to intermediate value and intermediate value to initial value can be expressed as Equation (1).

$$\frac{P_E^* - P_E^0}{P_D^{top} - P_E^*} \tag{1}$$

At this time, if the error rate between the various production links within the economy can be further reduced, then $\frac{P_E^*}{P_D^{top}}$ is reduced. In the digitization of the manufacturing industry, this reflection can manifest in applying digital technology to make the production process closer to the core technology while improving the modularity of the product and enhancing the relative height of the embedding position of the GVC.

In addition, the digital economy impacts the relative breadth of the GVC by promoting technological innovation. The digital economy can encourage the improvement of the technical innovation level of enterprises. For the manufacturing industry, the increased input in the digital economy represents the increase in investment in data assets and the broader application of digital technology, which has a significant spillover effect on manufacturing innovation. From the production side, the investment in digital technology can effectively shorten the R&D cycle, thus promoting the improvement of internal R&D efficiency and R&D capabilities. At the same time, through the extensive use of "big data", "blockchain", and "cloud computing digital communication technology", the ability of enterprises to integrate the external environment is enhanced, the cooperation and integration of various production processes is promoted, and the technical level of enterprises is improved.

On the basis of the above analysis, the following hypotheses can be proposed:

**H2:** *The input level of the digital economy in manufacturing affects the embedded position of the GVC by improving innovation efficiency.*

**H2.1:** *Improving the input level of the digital economy in the manufacturing industry enhances the relative breadth of the embedded position of the GVC by improving innovation efficiency.*

**H2.2:** *Improving the digital level of the manufacturing industry enhances the relative height of the embedded position of the GVC by improving innovation efficiency.*

*3.3. Improve the Efficiency of the Capital Allocation Mechanism*

3.3.1. Reduce Production Resource Allocation Costs

The reduction in the allocation cost of production resources, from the perspective of input and output, can be manifested as the reduction in the trade cost of intermediate goods and final goods, which can obtain higher trade-added value, thus showing the improvement of the embedding position of GVC [57].

The continuous penetration and deep integration of digital technology into various fields of the national economy, especially the effective integration of the manufacturing industry, can optimize the production structure, enhance sensitivity and flexibility to market changes, and improve the efficiency of resource search. The specialization of the social division of labor and the deepening of inter-industry linkage effects will inevitably lead to the rapid expansion of the number, variety, and scale of commodity exchanges between different producers, which will lead to an increase in various transaction costs [58]. The improvement of digital technology use in the manufacturing industry optimizes the efficiency of information transmission. The application of digital technology and the advancement of digital transmission efficiency assume the responsibility of some intermediary services so that the auxiliary services needed in the production process can be quickly matched through digital technology and digital information. The transaction costs in the manufacturing industry can be reduced in production transactions, thereby reducing the allocation costs of production resources.

On the other hand, improving the digital level of the manufacturing industry can help to enhance and strengthen the development of the digital economy and related sectors themselves, to achieve the scale of the digital industry, thus forming a scale effect. At the same time, from the perspective of production costs, the manufacturing industry digitizes some industries in the production sector, which is conducive to improving the level of enterprise specialization. Moreover, it reduces transaction costs by promoting the formation of scale effects. The simultaneous impact of the two aspects further reduces asset allocation costs.

### 3.3.2. Improve the Utilization Efficiency of Production Resources

In the stage of resource utilization, the improvement of the input level of the digital economy in manufacturing is conducive to deepening the specialization within the value chain. It is an essential driving force to promote the relative breadth of the embedded position of the GVC in manufacturing. Meanwhile, in the GVC division of labor, with the free flow and allocation of production factors, when the efficiency of resource utilization is improved, its production process can participate in more division of labor processes on the basis of its efficiency advantages, thus promoting the proportion of the economy in the GVC.

Improving the input level of the digital economy in manufacturing can promote the utilization efficiency of its production resources. In manufacturing resource utilization, through the broad application of data elements and digital technology, data can become a key factor of production and directly participate in show. Relying on open sharing and efficient use of data resources can promote the manufacturing industry to obtain market consumption trends in the production process, thereby improving the utilization efficiency of production resources. On the other hand, in the GVC division of labor, the professional division of labor level has been further enhanced. The development of the digital economy can promote the flow efficiency between resources, enhance the cooperation between enterprises, and further promote the development of the professional division of labor. With the help of the spillover effect of digital economy development in resource allocation, it promotes the improvement of resource utilization efficiency.

On the basis of the above analysis and theoretical mechanism, the following hypotheses can be proposed:

**H3:** *The improvement of the input level of the digital economy in manufacturing promotes the embedded position of the GVC to a high level by improving the efficiency of asset allocation.*

**H3.1:** *The improvement of the input level of the digital economy in manufacturing enhances the relative breadth of the embedded position of the GVC by improving the efficiency of asset allocation.*

## 4. Model Settings and Sample Selection

### 4.1. Model Settings

On the basis of the above analysis, a panel regression model is established to test the research hypotheses Empirically, the regression model is shown in Equations (2) and (3).

$$GVCpat_{it} = \alpha_1 + \beta_1 DIG\_T_{it} + \gamma_1 Control_{it} + \varepsilon_{it} \tag{2}$$

$$GVCps_{it} = \alpha_2 + \beta_2 DIG\_T_{it} + \gamma_2 Control_{it} + \varepsilon_{it} \tag{3}$$

where $GVCpat_{it}$ represents the relative breadth of the GVC embedding position, $GVCps_{it}$ represents the relative height of the GVC embedding position, $Control_{it}$ represents the control variable, $\varepsilon_{it}$ denotes the residual error, $i$ represents the industry, and $t$ represents the year.

### 4.2. Selection of Variable

### 4.2.1. Explained Variables

Relative breadth of the embedded position of the GVC: The relative breadth of a country's embeddedness position in the GVC describes the degree of correlation between the country's industry and the GVC. When a country participates in the division of labor in the GVC, the degree of correlation is represented by the number of intermediate goods produced for the country that become the finished goods of other countries and their value growth, accounting for the proportion of its total growth. The specific decomposition method is to create the GVC production decomposition model and decompose the added value of production in two directions: direction (forward) and source (backward) [59]. The formulas are shown in Equations (4) and (5).

$$Va' = \underbrace{\hat{V}LY^d}_{V\_D} + \underbrace{\hat{V}LY^F}_{V\_RT} + \underbrace{\hat{V}LA^F LY^d}_{V\_GVC\_S} + \underbrace{\hat{V}LA^F(B\hat{Y} - Y^d)}_{V\_GVC\_C} \tag{4}$$

$$Y' = \underbrace{\hat{V}LY^d}_{Y\_D} + \underbrace{\hat{V}LY^F}_{Y\_RT} + \underbrace{\hat{V}LA^F LY^d}_{Y\_GVC\_S} + \underbrace{\hat{V}LA^F(B\hat{Y} - Y^d)}_{Y\_GVC\_C} \tag{5}$$

where $\hat{V}$ represents the diagonal matrix of the input–output value-added coefficient, $L = (I - A^D)^{-1}$ is the Leontief inverse matrix, $V\_D$ and $Y\_D$ represent the added value of domestic production for domestic consumption, $V\_RT$ and $Y\_RT$ represent the value added of domestic production for export, becoming the intermediate goods of other countries, $V\_GVC\_S$ and $Y\_GVC\_S$ represent the domestic value added in the intermediate products exported by a country, which are absorbed by the direct importing country and processed into final products for consumption, $V\_GVC\_C$ and $Y\_GVC\_C$ are the added value of production undergoing more than two cross-border productions, $V\_GVC\_C$ represents the added value of domestic intermediate exports processed by the importing country and recorded as exports, and $Y\_GVC\_C$ represents the import value added of a country's export varieties. The forward and backward participation indices can be calculated through Equations (6) and (7).

$$GVCpt_F = \frac{V\_GVC\_S + V\_GVC\_C}{Va'} \tag{6}$$

$$GVCpt_B = \frac{Y\_GVC\_S + Y\_GVC\_C}{Y'} \tag{7}$$

where $GVCpt_F$ represents the GVC forward participation index, and $GVCpt_B$ represents the GVC backward participation index. The ratio of forward and backward participation can be further processed as the breadth of the GVC.

$$GVCpat = \frac{GVCpt_F}{GVCpt_B} \tag{8}$$

The relative height of the embedded position of the GVC: The size of the value chain is the number of times a value increase in the production process is recorded as a product from the initial input to the final product. This indicator can better reflect the complexity of the country's output products in the international production process. In the specific calculation, this paper chooses the method of WWYZ [60] to calculate the index. According to the number of cross-border transactions of intermediate products, this method classifies a single cross-border trade into a simple value chain and multiple cross-border transactions into a complex value chain. The calculation formulas are shown in Equations (9) and (10).

$$PLv\_GVC = PLv\_GVC\_S + PLv\_GVC\_C = \frac{Xv\_GVC\_S}{V\_GVC\_S} + \frac{Xv\_GVC\_C}{V\_GVC\_C} \tag{9}$$

$$PLy\_GVC = PLv\_GVC\_S + PLv\_GVC\_C = \frac{Xv\_GVC\_S}{V\_GVC\_S} + \frac{Xv\_GVC\_C}{V\_GVC\_C} \tag{10}$$

where $PLv\_GVC$, $PLv_{GVC_S}$, and $PLv\_GVC\_C$ respectively represent the lengths of the value chain, simple value chain, and complex value chain in forward decomposition, $Xv\_GVC\_S$ and $Xv\_GVC\_C$ represent the total output caused by domestic added value in the simple value chain and complex value chain, respectively, and $V\_GVC\_S$ and $V\_GVC\_C$ respectively represent the domestic added value in the simple value chain and complex value chain in forward decomposition.

Furthermore, the distance between the value generated in the production process and the final demand is called the upstream degree, and the distance from the initial input is called the downstream degree. Suppose the industry is close to the initial input and far from the final demand, which means the length of the forward value chain is greater than that of the backward value chain; this means that the position of the industry is upstream. On the other hand, if the drive is far from the initial input and close to the final demand, i.e., the length of the forward value chain is less than the length of the backward value chain, this means that the position of the industry is downstream. By measuring the relative part of the industrial value chain, the sector's height in the GVC can be expressed by the ratio of forward participation ($PLv$) to the backward involvement ($Ply$) of the value chain. The specific calculation formula is shown in Equation (11).

$$GVC_{ps} = \frac{PLv\_GVC}{PLy\_GVC} \tag{11}$$

### 4.2.2. Explanatory Variable

The input level of the digital economy in the manufacturing industry is the proportion of intermediate products used in the primary departments of the digital economy in the final finished products. In the industry selection of the digital economy, starting from the theoretical connotation and according to the classification standard of the industry classification of the fourth edition of the International Standard Industrial Classification of All Economic Activities (hereinafter referred to as ICIS Rev4.0), the primary departments of the digital economy include computer, electronic, and optical products manufacturing, postal service, communication, and telecommunications. The calculated method of the absolute and relative indices is used to determine the digital economy's input level [24,61].

First of all, the total consumption coefficient can fully reflect the overall impact of the digital industry on various manufacturing industries through the industrial correlation effect, and the formula is shown in Equation (12).

$$DC_{dj} = a_{dj} + \sum_{m=1}^{N} a_{dm}a_{mj} + \sum_{l=1}^{N}\sum_{m=1}^{N} a_{dl}a_{lm}a_{mj}\cdots \tag{12}$$

where the first item on the right side of the formula is the direct consumption of department *J* to department *D*, the second item is the first indirect consumption of department *J* to department *D* through department *M*, the third item is the second indirect consumption of department *J* to department *D* through departments *M* and *L*, etc. Equation (13) obtains the matrix of the complete consumption coefficient.

$$DCc = D + D2 + \cdots + Dm = (E - D) - 1 - E. \tag{13}$$

Furthermore, the direct input level of the digital economy $DIG\_D_{it}$ represents the ratio of digital input directly consumed by the manufacturing industry to the total. The formula is shown in Equation (14).

$$DIG\_D_{it} = \frac{\sum_d a_{dj}}{\sum_k a_{kj}} \tag{14}$$

where $a_{dj}$ is the direct consumption coefficient of digital economy department *D* in *J* department, and $a_{kj}$ is the direct consumption coefficient of department *J* from department *K*. On this basis, it can be expressed by total dependence ($DIG\_T_{jt}^c$), and its formula is shown in Equation (15).

$$DIG\_T_{jt}^c = \sum_d \left( complete_{dj} / \sum_{k=1}^{N} complete_{kj} \right) \tag{15}$$

where $complete_{kj}$, the complete consumption coefficient, shows the consumption from department *J* to each input in department K. The index of total dependence reveals the direct and indirect connection between the manufacturing and digital departments more thoroughly. It reflects the relative level of digital elements in all inputs more accurately.

### 4.2.3. Control Variable

In addition to the main explanatory variables, primary reference is made to the following factors on the embedded position of the GVC: (1) labor factor level (loba), described by the number of employees in the industry with a junior high school degree or above; (2) technology development level (tech), describing the R&D internal input operating income ratio of manufacturing industrial enterprises above the designated size; (3) industry import and export data (exp), using the export delivery value of manufacturing above the designated size as a characterization index; (4) industry development scale (scale), described by the operating income of industrial enterprises above the designated size because the industry development scale data are not easily obtained directly; (5) investment level (inve), described by the fixed asset investment index of the manufacturing industry; (6) production factor endowment (endow), described by the ratio of the industry's net value of fixed assets to the average number of employees.

### 4.3. Data Source and Description
#### 4.3.1. Industry Matching

The industries included in the manufacturing industry are defined by the standard of ICIS Rev4.0. Some of the data in the study came from the National Bureau of Statistics of China, where the industry classification is different from ICIS Rev4.0; thus, the respective industries were adjusted. In the actual calculation, the classification of "other transportation equipment manufacturing" was removed, and other industries were merged according

to the categories. Finally, 16 manufacturing industries were selected for analysis. The corresponding relationship of specific industries is shown in Table 1.

**Table 1.** Industry classification.

| Industry Code | ICIS Rev4.0 | National Economic Industry Classification |
|---|---|---|
| D10T12 | Food, beverages, and tobacco manufacturing | Agricultural and sideline food processing; food manufacturing; wine, materials, and refined tea manufacturing; tobacco manufacturing |
| D13T15 | Textiles, textile products, leather, and footwear manufacturing | Textile, garment, and clothing industry; leather, fur, feathers, and their products; footwear manufacturing |
| D16 | Wood and related products manufacturing | Wood processing; wood, bamboo, rattan, brown, and grass products industry; furniture manufacturing |
| D17T18 | Paper products and printed matter manufacturing | Papermaking and paper products industry; printing and recording media reproduction industry; cultural and educational, industrial, sports, and entertainment products manufacturing industry |
| D19 | Coke, oil refining, and nuclear fuel manufacturing | Coke, oil refining, and nuclear fuel manufacturing |
| D20 | Chemical raw materials and chemical products manufacturing | Chemical raw materials and chemical products manufacturing |
| D21 | Pharmaceutical products and chemical fiber products manufacturing | Pharmaceutical manufacturing and chemical fiber manufacturing |
| D22 | Rubber products and plastic manufacturing | Rubber products and plastic manufacturing |
| D23 | Non-metallic mineral products manufacturing | Non-metallic mineral products manufacturing |
| D24 | Basic metal manufacturing | Ferrous metal smelting and rolling processing industry; non-ferrous metal smelting and rolling processing industry |
| D25 | Metal products industry | Metal products industry |
| D26 | Computer, electronics, and optical equipment manufacturing | Computer, communications, and other electronic equipment manufacturing; instrument and meter manufacturing |
| D27 | Electrical machinery and equipment manufacturing industry | Electrical machinery and equipment manufacturing industry |
| D28 | Machinery and equipment manufacturing industry | General equipment manufacturing industry; special equipment manufacturing industry |
| D29 | Automobile manufacturing industry | Automobile manufacturing industry |
| D30 | Railway, ship, and other transportation equipment manufacturing industry | Railway, shipbuilding, aerospace, and other transportation equipment manufacturing |
| D31T33 | Other manufacturing | Other manufacturing |

4.3.2. Data Sources

As a function of the research needs and data availability, the observation period of this paper was from 2010 to 2018. Herein, the explained variable came from the UIBE database of the GVC Research Institute of the University of International Business and Economics; the explanatory variable data were mainly derived from the OECD input–output table (OECD-ICIO table) and calculated using the above-described method; the primary data of the control variables were derived from the "China Industrial Statistics Yearbook" and the National Bureau of Statistics of China.

### 4.3.3. Descriptive Statistics

Descriptive statistics of variables. The control variables were logarithmically processed to reduce the interference of data dimension. Table 2 presents the descriptive statistical results, revealing no outliers.

**Table 2.** Descriptive statistics of variables.

| Variable Name | Variable Interpretation | Number | Means | St.d | Min | Max |
|---|---|---|---|---|---|---|
| GVC_Ps | The relative height of the embedding position of the GVC | 144 | 0.8613 | 0.117 | 0.651 | 1.283 |
| GVC_Pat | The relative breadth of the embedding position of the GVC | 144 | 0.813 | 0.120 | 0.511 | 1.309 |
| DDIG_T | The input level of manufacturing digital economy | 144 | 0.038 | 0.066 | 0.009 | 0.334 |
| Llabo | Industry labor elements | 144 | 8.217 | 0.819 | 5.635 | 9.486 |
| Ltech | Industry technology level | 144 | 4.246 | 0.850 | −2.162 | 5.442 |
| Lexd | Industry trade level | 144 | 8.101 | 1.064 | 5.869 | 10.804 |
| Lscal | Industry scale | 144 | 10.526 | 0.923 | 7.424 | 11.741 |
| Linve | Industry investment level | 144 | 8.9458 | 0.740 | 7.006 | 11.595 |
| Lendow | Factor endowments of production | 144 | 3.3697 | 1.101 | 1.357 | 9.8362 |

## 5. Empirical Results and Analysis

### 5.1. Analysis of the Impact of the Digital Economy on the Promotion of the GVC

Firstly, the relative height of the embedded position of the GVC was analyzed. Observing the regression results in column 1 reveals that, with the improvement of the input level of the digital economy in the manufacturing industry, the relative height of the embedded position of the GVC was improved. The regression coefficient was 0.4119, and this result was significant at 90%. At the same time, the regression analysis of the input level of the digital economy in manufacturing and the relative breadth of the GVC embedding position showed a positive impact. The regression coefficient was 0.5568, and this result was significant at 90%. With the regression result, the impact of the input level of digital economics on the embedded position of the GVC was confirmed. The result of the basic regression is shown in Table 3.

**Table 3.** Result of the basic regression.

| | GVC_Pat | GVC_Ps |
|---|---|---|
| DIG_T | 0.5568 * | 0.4119 ** |
| | (0.3332) | (0.2047) |
| Llabo | −0.0465 * | −0.0170 |
| | (0.0274) | (0.0168) |
| Ltech | −0.0172 ** | −0.0068 ** |
| | (0.0052) | (0.0032) |
| Lexd | −0.0287 ** | −0.0125 |
| | (0.0138) | (0.0085) |
| Lscal | 0.0162 | 0.0362 ** |
| | (0.0267) | (0.0164) |
| Linve | 0.0202 | 0.0107 |
| | (0.0127) | (0.0078) |
| Lendow | 0.0088 * | 0.0072 ** |
| | (0.0051) | (0.0031) |
| _cons | 1.1168 *** | 0.6518 *** |
| | (0.2279) | (0.1400) |
| Time effect | fixed | fixed |
| Identify effect | fixed | fixed |
| N | 144 | 144 |
| $R^2$ | 0.356 | 0.572 |
| F-value | 4.17 | 10.07 |

Standard errors are in parentheses. * $p < 0.10$, ** $p < 0.05$, *** $p < 0.001$.

According to the above analysis, improving the input level of the digital economy in manufacturing plays a significant role in promoting the position of GVC embeddedness, which can promote the relative height and breadth of the position of GVC embeddedness. The above regression analysis can verify research hypothesis 1.

*5.2. Robustness Test*

On the basis of the above analysis, a robustness test was carried out on the benchmark regression to ensure the robustness of the research.

Firstly, the direct consumption coefficient of the digital economy was used as the replacement explanatory variable for regression analysis. The regression model is shown in Equations (16) and (17).

$$GVCpat_{it} = \alpha_1 + \beta_1 DIG\_D_{it} + \gamma_1 Control_{it} + \varepsilon_{it} \tag{16}$$

$$GVCps_{it} = \alpha_2 + \beta_2 DIG\_D_{it} + \gamma_2 Control_{it} + \varepsilon_{it} \tag{17}$$

where DIG_D represents the direct consumption coefficient of the input level of digital economics in manufacturing, with other variables remaining the same as the baseline regression. The regression results are shown in columns 1 and 2 in Table 4. The regression results are consistent with the baseline regression, indicating robust results.

**Table 4.** Test result of robustness.

|  | Change Variety | | 2SLS | |
|---|---|---|---|---|
|  | **GVC_Pat** | **GVC_Ps** | **GVC_Pat** | **GVC_Ps** |
| DIG_D | 0.8420 ** | 0.4895 ** |  |  |
|  | (0.2811) | (0.1745) |  |  |
| DIG_T |  |  | 0.9463 ** | 0.5858 ** |
|  |  |  | (0.3116) | (0.1909) |
| Llabo | −0.0532 ** | −0.0198 | −0.0516 ** | −0.0193 |
|  | (0.0267) | (0.0166) | (0.0244) | (0.0150) |
| Ltech | −0.0167 ** | −0.0066 ** | −0.0167 *** | −0.0066 ** |
|  | (0.0051) | (0.0031) | (0.0046) | (0.0028) |
| Lexd | −0.0286 ** | −0.0123 | −0.0293 ** | −0.0128 * |
|  | (0.0134) | (0.0083) | (0.0123) | (0.0075) |
| Lscal | 0.0260 | 0.0401 ** | 0.0245 | 0.0399 ** |
|  | (0.0261) | (0.0162) | (0.0239) | (0.0146) |
| Linve | 0.0157 | 0.0084 | 0.0190 * | 0.0102 |
|  | (0.0125) | (0.0078) | (0.0113) | (0.0069) |
| Lendow | 0.0090 * | 0.0073 ** | 0.0088 * | 0.0072 ** |
|  | (0.0050) | (0.0031) | (0.0046) | (0.0028) |
| _cons | 1.0892 *** | 0.6457 *** | 0.9563 *** | 0.6701 *** |
|  | (0.2211) | (0.1372) | (0.2202) | (0.1350) |
| Time effect | fixed | fixed | fixed | fixed |
| Identify effect | fixed | fixed | fixed | fixed |
| N | 144 | 144 | 144 | 144 |
| $R^2$ | 0.389 | 0.586 | 0.985 | 0.512 |
| F-value | 4.73 | 8.94 | - | - |

Standard errors are in parentheses. * $p < 0.10$, ** $p < 0.05$, *** $p < 0.001$.

Next, considering the endogeneity problem in the regression test, the instrumental variable method was used to reduce the endogeneity problem of the model. The input digitization degree of India's manufacturing industry was selected as the instrumental variable of China's manufacturing input digitization. On the one hand, both India and China occupy an essential position in the world manufacturing market. The economic foundations of the two countries are similar and influence each other, meeting the correlation requirements of instrumental variables; on the other hand, the digitalization level of

India's manufacturing is different, wherein promoting the embedded position of the GVC has little impact and can meet the exogenous demand.

To sum up, we used the development level of India's digital economy as an instrumental variable to control endogeneity. The digitization level of India's manufacturing industry was described by measuring the direct consumption coefficient of India's manufacturing and digital economy industries and calculating the degree of direct dependence. The data came from the OECD input–output table.

The 2SLS method was used for testing; the regression results are shown in columns 3 and 4 of Table 4. According to the weak instrumental variable test, the F value was greater than 10, and the *p*-value was 0.000. Thus, there was no invalid instrumental variable problem, and the regression results were significant at the 5% level. In summary, the instrumental variables were reasonable and adequate, and after overcoming potential endogeneity, the core conclusions of this paper were still robust. The result is shown in Table 4.

*5.3. Heterogeneity Test*

The heterogeneity of different industries was tested. In manufacturing production, according to the characteristics of its primary means of production and production forms, the manufacturing industry can generally be decomposed into labor-intensive, capital-intensive, and technology-intensive.

According to the regression results, the impact of the input level of the digital economy in manufacturing on the embedding position of the GVC has industry heterogeneity. The effect of digital economy input level on the GVC embedding position in the manufacturing industry is not significant in labor-intensive industries; for capital-intensive industries, with the increase in the level of investment in the digital economy, there is a negative impact on the relative breadth and relative height of the embedded position of the GVC. The impact coefficients were −6.7008 and −6.5032, respectively, and the results were significant at the 99% level. For capital-intensive industries, there is a significant positive impact on the relative height of the embedded position of the GVC; the coefficient was 0.3322, the result was significant at the 90% level. The impact on the relative breadth of the embedding position was not significant.

According to the regression results, the impact of the digital economy's input level on the embedded position of the GVC is insignificant, indicating that the effect has yet to appear in the sample time. For capital-intensive industries, the results are contrary to reality. Capital-intensive industries are mainly resource-dependent industries, such as the mining industry. For emerging economies such as China, improving the digital economy's input level means using advanced technology more widely. Meanwhile, there is an increase in the length of the production process. In addition, technical barriers are an influencing factor. Advanced technology has also increased the impact of technical barriers on related industries. For technology-intensive industries, the improvement of the input level of the digital economy has a significant effect on improving the relative height of the embedded position in the GVC, indicating that the innovation effect of the digital economy and the improvement of asset allocation efficiency promote the industry to move to a high-value acquisition position in the GVC. The result of heterogeneity regression is shown in Table 5.

**Table 5.** Result of the heterogeneity regression.

| | Labor-Intensive | | Capital-Intensive | | Technology Intensive | |
|---|---|---|---|---|---|---|
| | **GVC_Pat** | **GVC_Ps** | **GVC_Ps** | **GVC_Pat** | **GVC_Ps** | **GVC_Pat** |
| DDIG_T | −1.1927 | −0.4186 | −6.7008 ** | −6.5032 ** | 0.3322 * | 0.3339 |
| | (2.8069) | (1.0895) | (1.9167) | (2.4970) | (0.1754) | (0.2700) |
| Llabo | −0.0481 | −0.0128 | −0.1422 *** | −0.1863 *** | −0.0246 | −0.0831 |
| | (0.0531) | (0.0206) | (0.0365) | (0.0476) | (0.0331) | (0.0510) |

**Table 5.** *Cont.*

| | Labor-Intensive | | Capital-Intensive | | Technology Intensive | |
|---|---|---|---|---|---|---|
| | **GVC_Pat** | **GVC_Ps** | **GVC_Ps** | **GVC_Pat** | **GVC_Ps** | **GVC_Pat** |
| Ltech | −0.0133 ** | −0.0060 ** | 0.0348 | 0.0505 | 0.0043 | −0.0179 |
| | (0.0043) | (0.0017) | (0.0300) | (0.0391) | (0.0166) | (0.0256) |
| Lexd | −0.0009 | 0.0095 | −0.0082 | −0.0573 ** | −0.0675 ** | −0.1130 ** |
| | (0.0169) | (0.0066) | (0.0213) | (0.0277) | (0.0307) | (0.0472) |
| Lscal | −0.0271 | −0.0258 | 0.1479 ** | 0.1713 ** | −0.0308 * | −0.0066 |
| | (0.0542) | (0.0210) | (0.0582) | (0.0759) | (0.0176) | (0.0271) |
| Linve | 0.0115 | −0.0041 | −0.0565 | −0.0517 | 0.0087 | 0.0647 * |
| | (0.0101) | (0.0039) | (0.0391) | (0.0509) | (0.0212) | (0.0326) |
| Lendow | 0.0036 | 0.0019 | 0.0064 * | 0.0057 | −0.0022 | −0.0020 |
| | (0.0058) | (0.0022) | (0.0032) | (0.0042) | (0.0026) | (0.0040) |
| _cons | 1.3155 *** | 1.1802 *** | 0.9991 *** | 1.4624 *** | 1.8107 *** | 1.9164 *** |
| | (0.2267) | (0.0880) | (0.2635) | (0.3432) | (0.2805) | (0.4318) |
| N | 36 | 36 | 54 | 54 | 54 | 54 |
| R2 | 0.504 | 0.763 | 0.532 | 0.492 | 0.440 | 0.228 |
| F-value | 3.64 | 11.52 | 5.68 | 6.65 | 4.60 | 1.73 |

Standard errors are in parentheses. * $p < 0.10$, ** $p < 0.05$, *** $p < 0.001$.

### *5.4. Mechanism Inspection*

5.4.1. Model Setting and Introduction of Intermediary Variables

The mechanism of the improvement of the input level of the digital economy in the manufacturing industry on the embedded position of the GVC was further analyzed. According to the theoretical analysis, the efficiency of asset allocation and the level of innovation were tested. The mediating effect model was constructed for testing [62]. The regression model is shown in Equations (17)–(19).

$$\text{GVC\_Pat}_{it} = \alpha_0 + \alpha_1 DIG\_T_{it} + \alpha_2 Control_{it} + \varepsilon_{it} \tag{18}$$

$$\text{M}_{it} = \gamma_0 + \gamma_1 DIG\_T_{it} + \gamma_2 Control_{it} + \varepsilon_{it} \tag{19}$$

$$GVC\_Pat_{it} = \omega_0 + \omega_1 DIG\_T_{it} + \omega_2 M_{it} + \omega_3 Control_{it} + \varepsilon_{it} \tag{20}$$

where M_{it} denotes intermediary variables: capital allocation efficiency and technological innovation transformation ability.

In the specific calculation, the technological innovation transformation ability is measured by the product of the number of patent applications and the proportion of intermediate investment in the industry [63]. The efficiency of capital allocation refers to the ratio of industry capital productivity to the average of all industries [64]. In the specific calculation, the technological innovation conversion ability ($Tica_{it}$) is measured by the product of the number of patent applications and the proportion of intermediate input in the industry. The regression model is shown in Equation (20).

$$Tica_{it} = \frac{patent\ applications_{it}}{industry\ intermediate\ input_{it}} \tag{21}$$

The capital allocation efficiency ($cae_{it}$) is measured by referring to the method, and the ratio of the capital productivity of the industry to the average capital productivity of all industries is used as the characterization index, in which the capital productivity of the industry is the ratio between the capital stock of the constant industry and the total output value of the industry. The perpetual inventory method calculates the industry capital stock [65] (Zhang, 2004). The specific calculation formula is shown in Equations (21) and (22).

$$Cae_{it} = \frac{\dfrac{kit_{it}}{industry_{it}}}{\dfrac{\sum_{i=1}^{i=n} kitit}{\sum_{i=1}^{i=n} industry_{it}}} \qquad (22)$$

$$kit_t = kit_{t-1} * \delta + (finv_t / ipi_t) \qquad (23)$$

The $\delta$ depreciation rate is 9.04%, $kit_t$ represents the capital depreciation rate, and 2003 is selected as the base period of the capital stock. When t = 1, $kit_1 = fix_{2003} \times 10\%$. Meanwhile, $finv_t$ is the fixed investment, and $ipi_t$ is the investment price index. The data were sourced from the National Bureau of Statistics of China.

5.4.2. Mediating Effect Analysis of the Relative Breadth of GVC Embedding Position

The digital economy impacts the relative height of the embedded position of the GVC by increasing the level of innovation and selecting innovation output as an intermediary variable to explore the impact. The intermediary variable of technological innovation capability was measured by the product of the number of patent applications and the proportion of intermediate investment in the input–output table [64] (Wan and Wang, 2019).

From the regression results, the first column shows the result of Formula 1, indicating that the input level of the digital economy in the manufacturing industry had a significant positive impact on the relative breadth of the embedded position of the GVC. The second column is the influence degree of digitalization level on technical intermediary variables, which was positive, with a coefficient of 1.3167, and the result was significant at a 95% level. After adding intermediate variables, the third column shows the relationship between the digital flower level and the embedded position of the GVC. The technological intermediary variable significantly affected the embedded position of the GVC at the level of 5%, with a coefficient of 0.127, but the result at the digital level was insignificant. After the Sobel test, the *p*-value was 0.0653, indicating a mediation effect. The result of regression is shown in Table 6. Through the development, research hypothesis 2.1 was verified.

**Table 6.** Mechanism test of relative breadth of the embedded position of the GVC.

| | Basic Regression | Innovation Mediation Effect | GVC_Pat Mediation Effect |
|---|---|---|---|
| | GVC_Pat | Tica | GVC_Pat |
| DIG_T | 0.5568 * | 1.3167 ** | 0.3895 |
| | (0.3332) | (0.6555) | (0.3298) |
| Tica | | | 0.1270 ** |
| | | | (0.0465) |
| Llabo | −0.0465 * | −0.0320 | −0.0424 |
| | (0.0274) | (0.0538) | (0.0266) |
| Ltech | −0.0172 ** | 0.0052 | −0.019 *** |
| | (0.0052) | (0.0102) | (0.0051) |
| Lexd | −0.0287 ** | 0.0034 | −0.0291 ** |
| | (0.0138) | (0.0271) | (0.0134) |
| Local | 0.0162 | 0.0883 * | 0.0050 |
| | (0.0267) | (0.0526) | (0.0263) |
| Live | 0.0202 | −0.0012 | 0.0204 |
| | (0.0127) | (0.0250) | (0.0124) |
| Lendow | 0.0088 * | −0.0027 | 0.0092 * |
| | (0.0051) | (0.0101) | (0.0050) |
| _cons | 1.1168 *** | −0.6387 | 1.1979 *** |
| | (0.2279) | (0.4483) | (0.2236) |
| N | 144 | 144 | 144 |
| R² | 0.356 | 0.283 | 0.397 |
| F-value | 4.17 | 2.97 | 4.60 |

Standard errors are in parentheses. * $p < 0.10$, ** $p < 0.05$, *** $p < 0.001$.

### 5.4.3. Mediating Effect Analysis of the Relative Height of GVC Embedding Position

According to the above analysis, the digital economy's input level impacts the embedded position of the GVC by improving the efficiency of asset allocation and innovation. The mediating effect of the relative height of the embedded GVC was tested. Table 7 shows the regression results. Benchmark regression is shown in column 1, and the second column shows the influence of the input level of the digital economy on innovation efficiency; with a regression coefficient of 1.3167, the result was significant at the level of 95%. After adding the innovation ability index, the regression results are shown in column 3. The influence coefficient of the input level of the digital economy on the height of the embedded position of the GVC was 0.3949, and the result was significant at a 90% level. However, the innovation ability result was not significant. After the Sobel test, the *p*-value was 0.0074, and the intermediary effect was significant. For the intermediary effect test of capital allocation efficiency, the results are shown in columns 4 and 5. The influence coefficient of the digitalization level on capital allocation efficiency was 0.6295, and the results were significant at a 95% level. After adding intermediary variables, the influence coefficient was 0.4291, and the effect was significant at a 95% level. However, the variable of capital allocation efficiency was not significant. The Sobel test *p*-value was 0.0046, and the intermediary effect was significant. According to the result, the input level of the digital economy affects the relative height of the embedded GVC by improving the efficiency of innovation and asset allocation. The regression result is shown in Table 7, and the research hypotheses 2.2 and 3.1 were verified.

**Table 7.** Mechanism test of the relative height of the embedded position of the GVC.

| | Basic Regression | Innovation Mediation Effect | Capital Mediation Effect | | GVC_Ps Mediation Effect |
|---|---|---|---|---|---|
| | GVC_Ps | Tica | Cae | GVC_Ps | GVC_Ps |
| DDIG_T3 | 0.4119 ** | 1.3167 ** | 0.6295 ** | 0.3949 * | 0.4291 ** |
| | (0.2047) | (0.6555) | (0.2693) | (0.2091) | (0.2109) |
| Tica | | | | 0.0129 | |
| | | | | (0.0295) | |
| Cae | | | | | 0.0081 |
| | | | | | (0.0224) |
| Llabo | −0.0170 | −0.0320 | −0.0857 ** | −0.0166 | −0.0181 |
| | (0.0168) | (0.0538) | (0499) | (0.0169) | (0.0171) |
| Ltech | −0.0068 ** | 0.0052 | −0.0016 | −0.0069 ** | −0.0070 ** |
| | (0.0032) | (0.0102) | (0.0148) | (0.0032) | (0.0033) |
| Lexd | −0.0125 | 0.0034 | −0.0799 ** | −0.0126 | −0.0120 |
| | (0.0085) | (0.0271) | (0.0319) | (0.0085) | (0.0086) |
| Local | 0.0362 ** | 0.0883 * | −0.0901 *** | 0.0351 ** | 0.0365 ** |
| | (0.0164) | (0.0526) | (0.0200) | (0.0167) | (0.0165) |
| Live | 0.0107 | −0.0012 | 0.2934 *** | 0.0108 | 0.0093 |
| | (0.0078) | (0.0250) | (0.0288) | (0.0078) | (0.0088) |
| Lendow | 0.0072 ** | −0.0027 | −0.0073 | 0.0073 ** | 0.0072 ** |
| | (0.0031) | (0.0101) | (0.0117) | (0.0032) | (0.0032) |
| _cons | 0.6518 *** | −0.6387 | −0.0146 | 0.6601 *** | 0.6657 *** |
| | (0.1400) | (0.4483) | (0.1643) | (0.1418) | (0.1457) |
| N | 144 | 144 | 144 | 144 | 144 |
| $R^2$ | 0.572 | 0.283 | 0.843 | 0.573 | 0.573 |
| F-value | 10.07 | 2.66 | 40.55 | 9.39 | 9.38 |

Standard errors are in parentheses. * $p < 0.10$, ** $p < 0.05$, *** $p < 0.001$.

### 5.5. Threshold Effect

When conducting a heterogeneity test, we found that, when the industry classification of the manufacturing industry is differentiated, the input level of the digital economy on the embedded position of the GVC is different in capital-intensive and technology-intensive industries. According to a comprehensive theoretical analysis, the degree of digital development in the manufacturing industry has other effects on the embedded position of the GVC, and there may be a threshold effect. In addition to the above discussion, the economic impact of digital technology depends on a sound digital ecosystem, which

is due to the industrial scale effect. Building an excellent digital ecology and industrial system requires a particular scale of capital and technical support [66]. In the process of digital industry investment, the manufacturing industry needs to cross the "digital divide" to better play the positive role of digital technology in the embedding position of the GVC. Therefore, a threshold effect test was further carried out.

Referring to the Hansen threshold regression model, the specific setting model of the threshold effect is shown in Equation (23).

$$GVC_{it} = \theta_1 + \theta_2 DIG_{it} \cdot I(thr_{it} < \gamma_1) + \theta_3 DIG_{Tit} \cdot I(\gamma_{it} \leq thr_1) + \theta_4 Countrol_{it} + \alpha_i + \varepsilon_{it} \tag{24}$$

where $GVC_{it}$ represents the explained variable, $DIG_{it}$ denotes the level of the digital economy, $thr_{it}$ represents the threshold variable, which is measured by the level of innovation and investment, $I(\cdot)$ represents the indicator function, $\gamma_{it}$ represents the threshold, $Control_{it}$ denotes a control variable, and $\varepsilon_{it}$ represents a random perturbation term.

The regression results are shown in Table 8. First of all, from the perspective of the technology level, there was a single threshold effect on the impact of the technology level on the relative height and breadth of the embedded position of the GVC. Specifically, the threshold value of technology level to relative breadth was 3.33, and the threshold value to the relative height was 3.09. The threshold effect of capital on the embedded position of the GVC was insignificant, but there were two thresholds for the value chain height at 7.96 and 7.82.

**Table 8.** Threshold effect test.

| Explained Variable | Threshold Number | Technical Threshold | | | Capital Threshold | | |
|---|---|---|---|---|---|---|---|
| | | Threshold Value | *p*-Value | BS Value | Threshold Value | *p*-Value | BS Value |
| GVC_Pat | Single threshold | 3.3317 | 0.0067 | 300 | 7.9560 | 0.1300 | 300 |
| | Double threshold | 4.4279 | 0.0933 | 300 | 8.5285 | 0.4900 | 300 |
| GVC_Ps | Single threshold | 3.0891 | 0.000 | 300 | 7.9560 | 0.0233 | 300 |
| | Double threshold | 2.8055 | 0.1267 | 300 | 7.8242 | 0.0267 | 300 |

First, the level of digitalization in the manufacturing industry had a threshold effect on the relative height and breadth of the embedded position of the GVC. For the relative breadth, before the technical level crossed the threshold, the level of digitalization had a positive impact, the coefficient was 1.5108, and the result was significant at the 0% level. On the other hand, it became negative when the first threshold was crossed, the coefficient was −1.0843, and the result was significant at the 5% level. After crossing the second threshold, this benefit became positive, the coefficient was 0.4706, and the result was significant at the 10% level. The input level of the digital economics had a significant effect on the value. The impact of chain status showed a "U"-shaped impact with the change of technology level. For the height of the embedded position of the GVC, there was a single threshold for the technical level. After the technical level crossed the threshold, the level of digitalization had a significant positive effect on the height of the GVC, with a coefficient of 0.4706, and the result was significant at the 90% level.

From the perspective of capital effect, when the capital level reached the first threshold, the digital level had a negative effect on the relative height of the GVC, with an influence coefficient of −2.2726, and the result was significant at the level of 99%. After exceeding the second threshold, it became a positive influence, with an influence coefficient of 1.3910, and the result was significant at 95%. This shows that, when capital investment was in intermediate, it had a negative impact. After crossing the threshold, the negative impact disappeared, showing a significant positive impact. The result of the threshold model estimation is shown in Table 9.

**Table 9.** Threshold model estimation result.

| | Technology Threshold | | Capital Threshold |
|---|---|---|---|
| | GVC_Pat | GVC_Ps | GVC_Ps |
| Llabo | −0.0493 ** | −0.0462 ** | −0.0913 *** |
| | (0.0230) | (0.0172) | (0.0234) |
| Ltech | −0.0099 ** | −0.0081 | −0.0207 *** |
| | (0.0050) | (0.0049) | (0.0049) |
| Lexd | −0.0067 | 0.0192 * | −0.0168 |
| | (0.0115) | (0.0098) | (0.0124) |
| Lscal | −0.0162 | −0.0154 | −0.0017 |
| | (0.0214) | (0.0167) | (0.0224) |
| Linve | 0.0105 | −0.0018 | 0.0140 |
| | (0.0095) | (0.0074) | (0.0098) |
| Lendow | 0.0028 | 0.0053 ** | 0.0024 |
| | (0.0025) | (0.0020) | (0.0026) |
| DDIG_T | 1.5108 * | 0.0138 | −0.2137 |
| (thr < qx1) | (0.9029) | (1.3445) | (0.5877) |
| DDIG_T | −1.0843 ** | | −2.2726 *** |
| (qx1 ≤ thr ≤ qx2) | (0.4953) | | (0.5021) |
| DDIG_T | 0.4706 * | 0.4706 * | 0.3010 |
| (thr > qx1) | (0.2631) | (0.2631) | (0.1967) |
| Con | 1.2701 *** | 1.3746 *** | 1.3910 *** |
| | (0.1063) | (0.1351) | (0.1006) |
| N | 144 | 144 | 144 |
| R2 | 0.414 | 0.355 | 0.437 |
| F-value | 9.33 | 7.29 | 10.26 |

Standard errors in parentheses * $p < 0.10$, ** $p < 0.05$, *** $p < 0.001$.

## 6. Conclusions and Suggestion

### 6.1. Research Result and Conclusions

With the wide application of digital technology, the input level of the digital economy in the manufacturing industry has been an essential factor affecting the division of labor position of the industry in the GVC and promoting the embedding of GVC to the high-value acquisition position. Compared with the existing research results, this paper verifies that the input level of the manufacturing digital economy promotes the breadth and depth of the value chain embedding position and expands the verification dimension; The path of influence is tested, which enriches the relevant research; While discussing the threshold effect, we find out the current situation of China's manufacturing industry in the threshold, to have a clearer understanding of the impact effect. The specific research results and conclusions are as follows:

(1) The level of digital input in the manufacturing industry positively impacts the embedded GVC, Manifested in the relatively high promotion and relatively vast expansion of the embedded GVC in the GVC division. At the same time, this effect is significantly enhanced when changing the measured variables of digital economy input in the manufacturing industry and using the manufacturing industry to analyze the proportion of direct use of intermediate products in the digital economy industry. Furthermore, the results are still significant after the 2SLS test using tool variables to control endogeneity. From the perspective of the impact effect, with the improvement of the digitalization level of the manufacturing industry, the breadth and depth of the embedded position of the value chain have been greatly improved. According to the empirical results, under the international background in recent years, improving the input level of China's manufacturing digital economy is an essential driving force to promote the embedding position of the international value chain.

(2) Differentiating manufacturing industries according to the means of production and performing regression analysis on different manufacturing industries, respectively, it is found that the input level of the digital economy has the most significant influence on the

relative height of GVC embedding position in technology-intensive industries; The impact on the relative height and breadth of the value chain embedding position of labor-intensive enterprises is not significant. There is apparent industry heterogeneity.

(3) Through the analysis of the intermediary effect, it is found that the improvement of the input level of manufacturing digital economy affects the relative breadth of GVC embedding position by improving the innovation level; By improving the level of innovation and the efficiency of capital allocation, the relative breadth and height of GVC embedding position are affected, and the intermediary effect is remarkable. Therefore, improving innovation efficiency and asset allocation efficiency is how the input level of the manufacturing digital economy affects the promotion of the GVC embedding position in China. The widespread use of digital technology and digital products can help innovation efficiency, optimize resource allocation efficiency, and then promote the manufacturing industry to achieve the goal of high-value acquisition in the international value chain division.

(4) Furthermore, the threshold effect of technology and capital in the manufacturing industry is tested, and it found that there are technology threshold effects and capital Al threshold effects. Among them, there is a double threshold for the relative breadth of the GVC embedding position by technical level and a single threshold for the relative depth; For the capital threshold effect, the relative depth of the embedded position of the value chain presents a double threshold. At the current stage of development in China, the sensitivity of different means of production-intensive industries to the input level of the digital economy is different, which is also reflected in the threshold effect. As far as the technical level is concerned, it presents a "U"-shaped trend for the relative breadth of the embedded position of the GVC and a single threshold effect for the relative height of The e GVC embedding position shows that only when the industrial technology level exceeds a particular critical value can the promotion of the input level of the digital economy be reflected. Currently, China's manufacturing industry still needs to cross the technical threshold to stimulate digital technology's promotion effect on the value chain's embedded position. Regarding the capital level, China's manufacturing industry is between the first and second thresholds. At this stage, the digital economy has not promoted the embedding position of GVC. In general, it is necessary to promote further the technical and capital levels of the manufacturing industry to promote the digital economy's input level.

*6.2. Recommendations and Prospects*

The digital economy has been a significant development direction in recent years and is also an essential factor affecting the GVC division of the manufacturing industry. Through the result of the research, this paper puts forward the following development suggestions for the development of the manufacturing industry.

Firstly, the level of investment in the digital economy is a crucial driving force for the upward movement of the GVC towards higher-value acquisition. In the process of manufacturing industry development, it is essential to enhance the utilization level of digital technologies and increase the intensity of using key digital economic technologies. There is a significant demand for improvement in the GVC division of labor for China's manufacturing industry. Therefore, it is crucial to further focus on the in-depth development of cutting-edge digital technologies such as artificial intelligence, blockchain, the Internet of Things, 5G, robotics, and data mining, and their widespread application in the manufacturing industry. This technology will help to advance the level of investment in the digital economy and enhance the manufacturing industry's voice and influence in the GVC. At the strategic level, it is crucial to strengthen the construction of digital economic infrastructure and promote the transformation of the network system.

Secondly, it is essential to establish the path through which the digital economy influences changes in the GVC's embeddedness. The utilization of digital technologies and the level of investment in the digital economy impact innovation output efficiency and capital allocation efficiency. It is crucial to strengthen innovation efficiency by promoting talent development and the establishment of innovation bases focused on digital technologies.

In optimizing capital allocation efficiency, there is a need to accelerate the application of standard digital technologies such as 5G, artificial intelligence, and the Internet of Things in optimizing industrial layouts. The application of technology will facilitate the creation of a digital ecosystem that integrates sensing, transmission, storage, computation, and processing at critical nodes of capital allocation.

Thirdly, it is vital to guide digital transformation in enterprises based on their industrial development characteristics and position them in the right direction. Promoting the development of technology and capital within enterprises is a critical prerequisite for leveraging the role of digital technologies in guiding the embedding of the GVC. In the digitalization process in enterprises, differential guidance should be provided based on their technology and capital levels. On the one hand, it is necessary to promote further increases in capital scale by fostering the collaborative development of upstream and downstream companies within the enterprise's supply chain. This will enable the identification of the distribution characteristics of the enterprise's own supply chain and expand the scale effect. On the other hand, there is a need to actively introduce advanced digital business concepts and technologies, such as consumer and partner collaboration in design, automation, intelligence, integrated production, platform-based operations, and terminal-oriented marketing. This will facilitate the enhancement of technological capabilities.

### 6.3. Research Limitations and Prospects

#### 6.3.1. Insufficient Research

Based on the existing research, this paper discusses the influence of manufacturing digitalization on the embedding position of the international value chain. Limited by the availability of data and the reliability of research methods, there are still some areas for improvement in this paper, which are embodied in the following aspects.

As for the research on the digitalization level of the manufacturing industry through the international input-output table, the measurement method needs to be more precise for the industry classification of the input-output table, and there may be room for further refinement of the measurement accuracy of the digitalization level. At the same time, except the perspective of the input level of the digital economy, the degree of government support, the degree of social use, and the degree of technological development are not within the scope of quantitative consideration.

In addition, this paper only studies the influence of the input level of the digital economy in individual economies. The input of the digital economy in the manufacturing industry is reflected in the proportion of domestic digital economy-related industrial intermediates used by the manufacturing industry, and in the perspective of economic globalization, there are close contacts between economies. In manufacturing digitalization, besides paying attention to the proportion of digital economy-related industries used within economies, input from other economies is also an essential factor.

#### 6.3.2. Research Prospect

With the improvement of the development level of the digital economy and its increasing proportion in the economies of all countries in the world, there is much room for expansion in related research fields. Specific can be further studied from the following aspects.

Further, strengthen the research on the index system of the input level of manufacturing digital economy, and incorporate factors such as government behavior, social residents' recognition, and technological development into the index system to comprehensively describe the development of the digital economy industry.

From the perspective of an open economy, using the international input-output table, this paper further depicts the difference in the influence of the input level of the digital economy industry from the inside and outside of the economy on the embedding position of GVC from the perspective of global trade, and further discusses the corresponding influence mechanism.

At the same time, with the rapid development of the digital industry, the improvement of digital investment level in the manufacturing industry has a vast impact on the development of the manufacturing industry. In addition to the macro-level research on the influence of GVC embedding position, it is also an important research direction to explore the influence of digital economy investment on supply chain resilience and green development of enterprises from the perspective of industry and enterprise behavior in the middle and micro perspectives.

**Author Contributions:** Project administration, G.R. and M.L.; Data collection, data analysis, sriting draft and final manuscript, G.R.; Funding acquisition, M.L. All authors have read and agreed to the published version of the manuscript.

**Funding:** This research is funded by the R&D Program of Beijing Municipal Education Commission (Grant No. KJZD20191000401). This research is also supported by Beijing Laboratory of National Economic Security Early-warning Engineering.

**Institutional Review Board Statement:** Not applicable.

**Informed Consent Statement:** Not applicable.

**Data Availability Statement:** The sample data are sourced from the "OECD-ICIO table" (https://www.oecd.org/sti/ind/inter-country-input-output-tables.htm, accessed on 3 November 2022), National Bureau of Statistics (https://data.stats.gov.cn, accessed on 17 December 2022) and UIBE GVC Database, (http://gvcdb.uibe.edu.cn, accessed on 12 April 2022).

**Conflicts of Interest:** The authors declare no conflict of interest.

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
