# Peer review of "Research on the Impact of the Input Level of Digital Economics in Chinese Manufacturing on the Embedded Position of the GVC"

_sustainability, doi:10.3390/su151612468_

Round 1
Reviewer 1 Report
The manuscript addresses a very interesting topic, taking in account the positive externalities generated by digital transition in the world economy. The ideas are well presented, scientifically argued and presented in a logical order.I recommend its publication after making some adjustments. The main adjustments I recommend are:
1. In the first section, the authors must present the main achievements of other countries regarding digital transition (for example the results obatin in European Union in this field - https://reform-support.ec.europa.eu/what-we-do/digital-transition_en and https://www.europarl.europa.eu/news/en/headlines/priorities/digital-transformation and https://commission.europa.eu/strategy-and-policy/priorities-2019-2024/europe-fit-digital-age_en#:~:text=Digital%20technology%20is%20changing%20people%E2%80%99s%20lives.%20The%20EU%E2%80%99s,is%20determined%20to%20make%20this%20Europe%27s%20%E2%80%9CDigital%20Decade%E2%80%9D.)
2. The authors should insert a literature review section in order to identify the research gap that was identified in the international literature and that justifies the realization of this study.
3. The authors should better justify the choice of the analysed period.
4. The authors should better justify the choice of method.
5. In the section 5. Empirical results and analysis, the authors must present the results of the study in the context of similar researches that confirm or not the own results. In this section, the authors must used supplimentary references in order to improve the discussions about the results. Some recommendations are
a) Kan, D., Lyu, L., Huang, W., & Yao, W. (2022). Digital economy and the upgrading of the global value chain of China’s service industry. Journal of Theoretical and Applied Electronic Commerce Research, 17(4), 1279-1296.
b) Meng, S., Yan, H., & Yu, J. (2022). Global Value Chain participation and green innovation: Evidence from Chinese listed firms. International Journal of Environmental Research and Public Health, 19(14), 8403.
c) Zhou, R., Tang, D., Da, D., Chen, W., Kong, L., & Boamah, V. (2022). Research on China’s manufacturing industry moving towards the middle and high-end of the GVC driven by digital economy. Sustainability, 14(13), 7717.
6. The authors must insert in the final section, considerations regarding limitations of the research and future directions for the research.
Author Response
First of all, thank you very much for your guidance and valuable suggestions on my article. If there are any inadequacies, I hope you will continue to make valuable suggestions.
Please see the attachment.

Reviewer 2 Report
Comments and Suggestions for Authors
This research presents the impact of digital input on the Chinese manufacture GVC degree. Where authors analyzes the influence of the input level of the digital economy on the embedded position of the GVC and draws the following conclusions:
- The level of digital input in the manufacturing industry positively impacts the embedded GVC, manifested in the relatively high promotion and relatively vast expansion of the embedded GVC in the GVC division.
- The promotion of digital input level in the manufacturing industry mainly affects the embedded position of the GVC through two channels. On the one hand, the relative height and breadth of the embedded position of the manufacturing industry in the GVC are improved by improving the efficiency of innovation output channels; On the other hand, by improving the efficiency of asset allocation, the relative breadth of the embedded GVC is promoted.
- Through the heterogeneity test, the influence of digital economy input on the embedded GVC varies with different manufacturing industries.
The article have a significant positive impact on practices to sustainability. The importance of this article is the threshold effect of technology and capital in the manufacturing industry is tested and found that there are technology threshold effects and capital threshold effects. Based on the technical threshold effect, it presents a "U"- shaped trend for the relative breadth of the embedded position of the GVC and a single threshold effect for the relative height of the GVC embedding position; Based on the capital threshold effect, it presents a double threshold for relative height. Overall, after the technical and capital levels of the industry have crossed the threshold, the input level of the digital economy has a more significant positive impact on the embedded GVC.
Through the result of the research, this paper puts forward the following development suggestions for the development of the manufacturing industry:
- the level of investment in the digital economy is a crucial driving force for the upward movement of the GVC towards higher-value acquisition.
- it is essential to establish the path through which the digital economy influences changes in the GVC's embeddedness. The utilization of digital technologies and the level of investment in the digital economy impact innovation output efficiency and capital allocation efficiency. It is crucial to strengthen innovation efficiency by promoting talent development and the establishment of innovation bases focused on digital technologies.
In general, the contents of this manuscript fully meet the requirements of scientific papers. The authors proposed an exciting method with many encouraging results. The article has a scientific style, and it sounds scientific.
Recommendations:
- Investigate in literature research what similar solutions exist internationally.
Author Response

(The authors gave the same response as above.)

Reviewer 3 Report
First of all, I would like to thank you for the possibility of reviewing this interesting paper that I have read with great interest.
The paper may have a clear interest associated to researchers from different scientific disciplines and, therefore, could have a notable repercussion in specialized scientific literature.
Why is this study necessary? should make clear arguments to explain what the originality and value of the proposed model is. This should be stated in the final paragraphs of introduction and conclusion sections.
Literature overview
I would like to suggest the following references:
Banga, K. (2022). Digital technologies and product upgrading in global value chains: Empirical evidence from Indian manufacturing firms. The European Journal of Development Research, 1-26.Blázquez, L., Díaz-Mora, C., & González-Díaz, B. (2023). Slowbalisation or a “New” type of GVC participation? The role of digital services. Journal of Industrial and Business Economics, 50(1), 121-147.
Conclusions: pleas add theoretical, managerial, and practical implications, limitation and further research. Some parts are included but must be extended.
However, I hope that all these comments will serve the author to improve the quality of the paper. Finally, I hope that the comments will be understood positively by the authors of this interesting paper.
Good luck!
Author Response
First of all, thank you very much for your guidance and valuable suggestions on my article. If there are any inadequacies, I hope you will continue to make helpful suggestions.
please see the attachment.

Reviewer 4 Report
Recommendations for the authors of the article:
1. It is necessary to correct the section of the article: "abstract". There is a lack of used research methods and research limitations (the authors also did not describe research limitations at the end). A description of the structure of the issues described in the article should also be added. Similar remarks should be taken into account in the introduction to the article.
2. Add a section: "Literature review".
3. The article lacks a description of the impact of the digital economy on sustainable development.
4. In the article the conclusions of the studies should be given in sub-paragraphs. This section requires a lot of substantive reinforcement.
Author Response
First of all, thank you very much for your guidance and valuable suggestions on my article. If there are any inadequacies, I hope you will continue to make helpful suggestions.
Please see the attachment.

Round 2
Reviewer 1 Report
The authors took into account the recommendations made by the reviewers and improved, quantitatively and qualitatively, the manuscript. Considering the additions brought by the authors, I believe that the manuscript can be published.
Reviewer 4 Report
Dear Authors, I think in this version the article is scientifically, methodologically and empirically on a good level. Congratulations. I wish you scientific and professional success.